# Continuous-time Value Function Approximation in Reproducing Kernel Hilbert Spaces

**Motoya Ohnishi**
Keio Univ., KTH, RIKEN
motoya.ohnishi@riken.jp

**Masahiro Yukawa**
Keio Univ., RIKEN
yukawa@elec.keio.ac.jp

**Mikael Johansson**
KTH
mikaelj@ee.kth.se

**Masashi Sugiyama**
RIKEN, Univ. Tokyo
masashi.sugiyama@riken.jp

## Abstract

Motivated by the success of reinforcement learning (RL) for discrete-time tasks such as AlphaGo and Atari games, there has been a recent surge of interest in using RL for continuous-time control of physical systems (cf. many challenging tasks in OpenAI Gym and DeepMind Control Suite). Since discretization of time is susceptible to error, it is methodologically more desirable to handle the system dynamics directly in continuous time. However, very few techniques exist for continuous-time RL and they lack flexibility in value function approximation. In this paper, we propose a novel framework for model-based continuous-time value function approximation in reproducing kernel Hilbert spaces. The resulting framework is so flexible that it can accommodate any kind of kernel-based approach, such as Gaussian processes and kernel adaptive filters, and it allows us to handle uncertainties and nonstationarity without prior knowledge about the environment or what basis functions to employ. We demonstrate the validity of the presented framework through experiments.

## 1 Introduction

Reinforcement learning (RL) [37, 20, 35] has been successful in a variety of applications such as AlphaGo and Atari games, particularly for discrete stochastic systems. Recently, application of RL to physical control tasks has also been gaining attention, because solving an optimal control problem (or the Hamilton-Jacobi-Bellman-Isaacs equation) [21] directly is computationally prohibitive for complex nonlinear system dynamics and/or cost functions.

In the physical world, states and actions are continuous, and many dynamical systems evolve in continuous time. OpenAI Gym [7] and DeepMind Control Suite [40] offer several representative examples of such physical tasks. When handling continuous-time (CT) systems, CT formulations are methodologically desirable over the use of discrete-time (DT) formulations with the small time intervals, since such discretization is susceptible to errors. In terms of computational complexities and the ease of analysis, CT formulations are also more advantageous over DT counterparts for control-theoretic analyses such as *stability* and *forward invariance* [14], which are useful for safety-critical applications. As we will show in this paper, our framework allows to constrain control inputs and/or states in a computationally efficient way.

One of the early examples of RL for CT systems [4] pointed out that Q learning is incabable of learning in continuous time and proposed advantage updating. Convergence proofs were given in [25] for systems described by stochastic differential equations (SDEs) [28] using a grid-based

Table 1: Relations to the existing approaches

|  | DT | DT stochastic (MDP) | CT | CT stochastic |
|---|---|---|---|---|
| Non kernel-based | (e.g. [5]) | (e.g. [44]) | (e.g. [8]) | (e.g. [25]) |
| **Kernel-based** | **(e.g. [9])** | **(e.g. [13])** | **(This work)** | **(This work)** |

discretization of states and time. Stochastic differential dynamic programming and RL have also been studied in, for example, [43, 30, 42]. For continuous states and actions, function approximators are often employed instead of finely discretizing the state space to avoid the explosion of computational complexities. The work in [8] presented an application of CT-RL by function approximators such as Gaussian networks with fixed number of basis functions. In [45], it was mentioned that any continuously differentiable value function (VF) can be approximated by increasing the number of independent basis functions to infinity in CT scenarios, and a CT policy iteration was proposed.

However, without resorting to the theory of reproducing kernels [3], determining the number of basis functions and selecting the suitable basis function class cannot be performed systematically in general. Nonparametric learning is often desirable when no *a priori* knowledge about a suitable set of basis functions for learning is available. Kernel-based methods have many non-parametric learning algorithms, ranging from Gaussian processes (GPs) [32] to kernel adaptive filters (KFs) [22], which can provably deal with uncertainties and nonstationarity. While DT kernel-based RL was studied in [29, 49, 41, 36, 26, 13, 27], for example, and the Gaussian process temporal difference (GPTD) algorithm was presented in [9], no CT kernel-based RL has been proposed to our knowledge. Moreover, there is no unified framework in which existing kernel methods and their convergence/tracking analyses are straightforwardly applied to model-based VF approximation.

In this paper, we present a novel theoretical framework of model-based CT-VF approximation in reproducing kernel Hilbert spaces (RKHSs) [3] for systems described by SDEs. The RKHS for learning is defined through one-to-one correspondence to a user-defined RKHS in which the VF being obtained is lying. We then obtain the associated kernel to be used for learning. The resulting framework renders any kind of kernel-based methods applicable in model-based CT-VF approximation, including GPs [32] and KFs [22]. In addition, we propose an efficient barrier-certified policy update for CT systems, which implicitly enforces state constraints. Relations of our framework to the existing approaches for DT, DT stochastic (the Markov decision process (MDP)), CT, and CT stochastic systems are shown in Table 1. Our proposed framework covers model-based VF approximation working in RKHSs, including those for CT and CT stochastic systems. We verify the validity of the framework on the classical Mountain Car problem and a simulated inverted pendulum.

## 2   Problem setting

Throughout, $\mathbb{R}$, $\mathbb{Z}_{\geq 0}$, and $\mathbb{Z}_{>0}$ are the sets of real numbers, nonnegative integers, and strictly positive integers, respectively. We suppose that the system dynamics described by the SDE [28],

$$dx = h(x(t), u(t))dt + \eta(x(t), u(t))dw, \tag{1}$$

is known or learned, where $x(t) \in \mathbb{R}^{n_x}$, $u(t) \in \mathcal{U} \subset \mathbb{R}^{n_u}$, and $w$ are the state, control, and a Brownian motion of dimensions $n_x \in \mathbb{Z}_{>0}$, $n_u \in \mathbb{Z}_{>0}$, and $n_w \in \mathbb{Z}_{>0}$, respectively, $h : \mathbb{R}^{n_x} \times \mathcal{U} \to \mathbb{R}^{n_x}$ is the drift, and $\eta : \mathbb{R}^{n_x} \times \mathcal{U} \to \mathbb{R}^{n_x \times n_w}$ is the diffusion. A Brownian motion can be considered as a process noise, and is known to satisfy the Markov property [28]. Given a policy $\phi : \mathbb{R}^{n_x} \to \mathcal{U}$, we define $h^\phi(x) := h(x, \phi(x))$ and $\eta^\phi(x) := \eta(x, \phi(x))$, and make the following two assumptions.

**Assumption 1.** *For any Lipschitz continuous policy $\phi$, both $h^\phi(x)$ and $\eta^\phi(x)$ are Lipschitz continuous, i.e., the stochastic process defined in (1) is an Itô diffusion [28, Definition 7.1.1], which has a pathwise unique solution for $t \in [0, \infty)$.*

**Assumption 2.** *The set $\mathcal{X} \subset \mathbb{R}^{n_x}$ is compact with nonempty interior $\mathrm{int}(\mathcal{X})$, and $\mathrm{int}(\mathcal{X})$ is invariant under the system (1) with any Lipschitz continuous policy $\phi$, i.e.,*

$$P_x(x(t) \in \mathrm{int}(\mathcal{X})) = 1, \ \forall x \in \mathrm{int}(\mathcal{X}), \ \forall t \geq 0, \tag{2}$$

*where $P_x(x(t) \in \mathrm{int}(\mathcal{X}))$ denotes the probability that $x(t)$ lies in $\mathrm{int}(\mathcal{X})$ when starting from $x(0) = x$.*

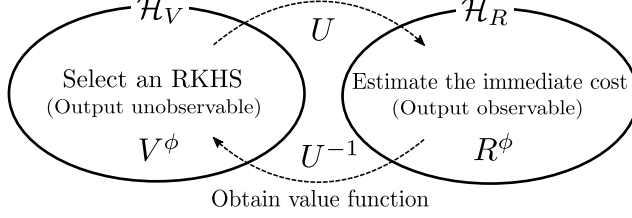

Figure 1: An illustration of the main ideas of our proposed framework. Given a system dynamics and an RKHS $\mathcal{H}_V$ for the VF $V^\phi$, define $\mathcal{H}_R$ under one-to-one correspondence to estimate an observable immediate cost function in $\mathcal{H}_R$, and obtain $V^\phi$ by bringing it back to $\mathcal{H}_V$.

Assumption 2 implies that a solution of the system (1) stays in $\mathrm{int}(\mathcal{X})$ with probability one. We refer the readers to [15] for stochastic stability and invariance for SDEs.

In this paper, we consider the immediate cost function[1] $R : \mathbb{R}^{n_x} \times \mathcal{U} \to \mathbb{R}$, which is continuous and satisfies $E_x \left[ \int_0^\infty e^{-\beta t} |R(x(t), u(t))| dt \right] < \infty$, where $E_x$ is the expectation for all trajectories (time evolutions of $x(t)$) starting from $x(0) = x$, and $\beta \geq 0$ is the discount factor. Note this boundedness implies that $\beta > 0$ or that there exists a zero-cost state which is stochastically asymptotically stable [15]. Specifically, we consider the case where the immediate cost is *not* known *a priori* but is sequentially observed. Now, the VF associated with a policy $\phi$ is given by

$$V^\phi(x) := E_x \left[ \int_0^\infty e^{-\beta t} R^\phi(x(t)) dt \right] < \infty, \tag{3}$$

where $R^\phi(x(t)) := R(x(t), \phi(x(t)))$.

The advantages of using CT formulations include a smooth control performance and an efficient policy update, and CT formulations require no elaborative partitioning of time [8]. In addition, our work shows that CT formulations make control-theoretic analyses easier and computationally more efficient and are more advantageous in terms of susceptibility to errors when the time interval is small. We mention that the CT formulation can still be considered in spite of the fact that the algorithm is implemented in discrete time.

With these problem settings in place, our goal is to estimate the CT-VF in an RKHS and improve policies. However, since the output $V^\phi(x)$ is unobservable and the so-called double-sampling problem exists when approximating VFs (see e.g., [38, 16]), kernel-based supervised learning and its analysis cannot be directly applied to VF approximation in general. Motivated by this fact, we propose a novel model-based CT-VF approximation framework which enables us to conduct kernel-based VF approximation as supervised learning.

## 3 Model-based CT-VF approximation in RKHSs

In this section, we briefly present an overview of our framework; We take the following steps:

1. Select an RKHS $\mathcal{H}_V$ which is supposed to contain $V^\phi$ as one of its elements.

2. Construct another RKHS $\mathcal{H}_R$ under one-to-one correspondence to $\mathcal{H}_V$ through a certain bijective linear operator $U : \mathcal{H}_V \to \mathcal{H}_R$ to be defined later in the next section.

3. Estimate the immediate cost function $R^\phi$ in the RKHS $\mathcal{H}_R$ by kernel-based supervised learning, and return its estimate $\hat{R}^\phi$.

4. An estimate of the VF $V^\phi$ is immediately obtained by $U^{-1}(\hat{R}^\phi)$.

An illustration of our framework is depicted in Figure 1. Note we can avoid the double-sampling problem because the operator $U$ is deterministic even though the system dynamics is stochastic. Therefore, under this framework, model-based CT-VF approximation in RKHSs can be derived, and convergence/tracking analyses of kernel-based supervised learning can also be applied to VF approximation.

**Algorithm 1** Model-based CT-VF Approximation in RKHSs with Barrier-Certified Policy Updates

> **Estimate of the VF:** $\hat{V}_n^\phi = U^{-1}(\hat{R}_n^\phi)$
> **for** $n \in \mathbb{Z}_{\geq 0}$ **do**
>     - Receive $x_n \in \mathcal{X}$, $\phi(x_n) \in \mathcal{U}$, and $R(x_n, \phi(x_n)) \in \mathbb{R}$
>     - Update the estimate $\hat{R}_n^\phi$ of $R^\phi$ by using some kernel-based method in $\mathcal{H}_R$    ▷ e.g., Section 6
>     - Update the policy with barrier certificates when $V^\phi$ is well estimated        ▷ e.g., (11)
> **end for**

**Policy update while restricting certain regions of the state space**    As mentioned above, one of the advantages of a CT framework is its affinity for control-theoretic analyses such as *stability* and *forward invariance*, which are useful for safety-critical applications. For example, suppose that we need to restrict the region of exploration in the state space to some set $\mathcal{C} := \{x \in \mathcal{X} \mid b(x) \geq 0\}$, where $b : \mathcal{X} \to \mathbb{R}$ is smooth. This is often required for safety-critical applications.

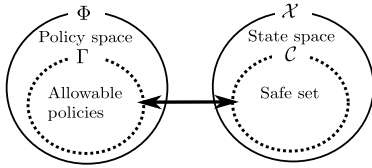

Figure 2: An illustration of barrier-certified policy updates. State constraints are implicitly enforced via barrier certificates.

To this end, control inputs must be properly constrained so that the state trajectory remains inside the set $\mathcal{C}$. In the safe RL context, there exists an idea of considering a smaller space of allowable policies (see [11] and references therein). To effectively constrain policies, we employ control barrier certificates (cf. [50, 48, 12, 46, 2, 1]). Without explicitly calculating the state trajectory over a long time horizon, it is known that any Lipschitz continuous policy satisfying control barrier certificates renders the set $\mathcal{C}$ forward invariant [50], i.e., the state trajectory remains inside the set $\mathcal{C}$. In other words, we can implicitly enforce state constraints by satisfying barrier certificates when updating policies. Barrier-certified policy update was first introduced in [27] for DT systems, but is computationally more efficient in our CT scenario. This concept is illustrated in Figure 2, where $\Phi$ is the space of Lipschitz continuous policies $\phi : \mathcal{X} \to \mathcal{U}$, and $\Gamma$ is the space of barrier-certified allowable policies.

A brief summary of the proposed model-based CT-VF approximation in RKHSs is given in Algorithm 1. In the next section, we present theoretical analyses of our framework.

## 4 Theoretical analyses

We presented the motivations and an overview of our framework in the previous section. In this section, we validate the proposed framework from theoretical viewpoints. Because the output $V^\phi(x)$ of the VF is unobservable, we follow the strategy presented in the previous section. First, by properly identifying the RKHS $\mathcal{H}_V$ which is supposed to contain the VF, we can implicitly restrict the class of the VF. If the VF $V^\phi$ is twice continuously differentiable[2] over $\text{int}(\mathcal{X}) \subset \mathcal{X}$, we obtain the following Hamilton-Jacobi-Bellman-Isaacs equation [28]:

$$\beta V^\phi(x) = -\mathcal{G}(V^\phi)(x) + R^\phi(x), \ x \in \text{int}(\mathcal{X}), \tag{4}$$

where the infinitesimal generator $\mathcal{G}$ is defined as

$$\mathcal{G}(V^\phi)(x) := -\frac{1}{2}\text{tr}\left[\frac{\partial^2 V^\phi(x)}{\partial x^2} A^\phi(x)\right] - \frac{\partial V^\phi(x)}{\partial x} h^\phi(x), \ x \in \text{int}(\mathcal{X}). \tag{5}$$

Here, $\text{tr}$ stands for the trace, and $A^\phi(x) := A(x, \phi(x)) \in \mathbb{R}^{n_x \times n_x}$, where $A(x, u) = \eta(x, u)\eta(x, u)^\mathsf{T}$. By employing a suitable RKHS such as a Gaussian RKHS for $\mathcal{H}_V$, we can guarantee twice continuous differentiability of an estimated VF. Note that functions in a Gaussian RKHS are smooth [23], and any continuous function on every compact subset of $\mathbb{R}^{n_x}$ can be approximated with an arbitrary accuracy [34] in a Gaussian RKHS.

Next, we need to construct another RKHS $\mathcal{H}_R$ which contains the immediate cost function $R^\phi$ as one of its element. The relation between the VF and the immediate cost function is given by rewriting (4) as

$$R^\phi(x) = [\beta I_{\text{op}} + \mathcal{G}] (V^\phi)(x), \ x \in \text{int}(\mathcal{X}), \tag{6}$$

where $I_{\text{op}}$ is the identity operator. To define the operator $\beta I_{\text{op}} + \mathcal{G}$ over the whole $\mathcal{X}$, we use the following definition.

**Definition 1** ([52, Definition 1])**.** *Let* $I_s := \{\alpha := [\alpha^1, \alpha^2, \ldots, \alpha^{n_x}]^{\mathsf{T}} \in \mathbb{Z}_{\geq 0}^{n_x} \mid \sum_{j=1}^{n_x} \alpha^j \leq s\}$ *for* $s \in \mathbb{Z}_{\geq 0}$, $n_x \in \mathbb{Z}_{>0}$. *Define* $D^\alpha \varphi(x) = \frac{\partial^{\sum_{j=1}^{n_x} \alpha^j}}{\partial (x^1)^{\alpha^1} \partial (x^2)^{\alpha^2} \ldots \partial (x^{n_x})^{\alpha^{n_x}}} \varphi(x)$, *where* $x := [x^1, x^2, \ldots, x^{n_x}]^{\mathsf{T}} \in \mathbb{R}^{n_x}$. *If* $\mathcal{X} \subset \mathbb{R}^{n_x}$ *is compact with nonempty interior* $\text{int}(\mathcal{X})$, $C^s(\text{int}(\mathcal{X}))$ *is the space of functions* $\varphi$ *over* $\text{int}(\mathcal{X})$ *such that* $D^\alpha \varphi$ *is well defined and continuous over* $\text{int}(\mathcal{X})$ *for each* $\alpha \in I_s$. *Define* $C^s(\mathcal{X})$ *to be the space of continuous functions* $\varphi$ *over* $\mathcal{X}$ *such that* $\varphi|_{\text{int}(\mathcal{X})} \in C^s(\text{int}(\mathcal{X}))$ *and that* $D^\alpha(\varphi|_{\text{int}(\mathcal{X})})$ *has a continuous extension* $D^\alpha \varphi$ *to* $\mathcal{X}$ *for each* $\alpha \in I_s$. *If* $\kappa \in C^{2s}(\mathcal{X} \times \mathcal{X})$, *define* $(D^\alpha \kappa)_x(y) = \frac{\partial^{\sum_{j=1}^{n_x} \alpha^j}}{\partial (x^1)^{\alpha^1} \partial (x^2)^{\alpha^2} \ldots \partial (x^{n_x})^{\alpha^{n_x}}} \kappa(y, x), \ \forall x, y \in \text{int}(\mathcal{X})$.

Now, suppose that $\mathcal{H}_V$ is an RKHS associated with the reproducing kernel $\kappa^V(\cdot, \cdot) \in C^{2 \times 2}(\mathcal{X} \times \mathcal{X})$. Then, we define the operator $U : \mathcal{H}_V \to \mathcal{H}_R := \{\varphi \mid \varphi(x) = U(\varphi^V)(x), \ \exists \varphi^V \in \mathcal{H}_V, \ \forall x \in \mathcal{X}\}$ as

$$U(\varphi^V)(x) := \beta \varphi^V(x) - [D^{e_1} \varphi^V(x), D^{e_2} \varphi^V(x), \ldots, D^{e_{n_x}} \varphi^V(x)] h^\phi(x)$$

$$- \frac{1}{2} \sum_{m,n=1}^{n_x} A_{m,n}^\phi(x) D^{e_m + e_n} \varphi^V(x), \qquad \forall \varphi^V \in \mathcal{H}_V, \ \forall x \in \mathcal{X}, \tag{7}$$

where $A_{m,n}^\phi(x)$ is the $(m, n)$ entry of $A^\phi(x)$. Note here that $U(\varphi^V)(x) = [\beta I_{\text{op}} + \mathcal{G}] (\varphi^V)(x)$ *over* $\text{int}(\mathcal{X})$. We emphasize here that the *expected* value and the immediate cost are related through the *deterministic* operator $U$. The following main theorem states that $\mathcal{H}_R$ is indeed an RKHS under Assumptions 1 and 2, and its corresponding reproducing kernel is obtained.

**Theorem 1.** *Under Assumptions 1 and 2, suppose that* $\mathcal{H}_V$ *is an RKHS associated with the reproducing kernel* $\kappa^V(\cdot, \cdot) \in C^{2 \times 2}(\mathcal{X} \times \mathcal{X})$. *Suppose also that (i)* $\beta > 0$, *or that (ii)* $\mathcal{H}_V$ *is a Gaussian RKHS, and there exists a point* $x_{t \to \infty} \in \text{int}(\mathcal{X})$ *which is stochastically asymptotically stable over* $\text{int}(\mathcal{X})$, *i.e.,* $P_x \left( \lim_{t \to \infty} x(t) = x_{t \to \infty} \right) = 1$ *for any starting point* $x \in \text{int}(\mathcal{X})$. *Then, the following statements hold.*
*(a) The space* $\mathcal{H}_R := \{\varphi \mid \varphi(x) = U(\varphi^V)(x), \ \exists \varphi^V \in \mathcal{H}_V, \ \forall x \in \mathcal{X}\}$ *is an isomorphic Hilbert space of* $\mathcal{H}_V$ *equipped with the inner product defined by*

$$\langle \varphi_1, \varphi_2 \rangle_{\mathcal{H}_R} := \langle \varphi_1^V, \varphi_2^V \rangle_{\mathcal{H}_V}, \ \varphi_i(x) := U(\varphi_i^V)(x), \ \forall x \in \mathcal{X}, \ i \in \{1, 2\}, \tag{8}$$

*where the operator* $U$ *is defined in (7).*
*(b) The Hilbert space* $\mathcal{H}_R$ *has the reproducing kernel given by*

$$\kappa(x, y) := U(K(\cdot, y))(x), \ x, y \in \mathcal{X}, \tag{9}$$

*where*

$$K(x, y) = \beta \kappa^V(x, y) - [(D^{e_1} \kappa^V)_y(x), (D^{e_2} \kappa^V)_y(x), \ldots, (D^{e_{n_x}} \kappa^V)_y(x)] h^\phi(y)$$

$$- \frac{1}{2} \sum_{m,n=1}^{n_x} A_{m,n}^\phi(y) (D^{e_m + e_n} \kappa^V)_y(x). \tag{10}$$

*Proof.* See Appendices A and B in the supplementary document. $\qquad\square$

Under Assumptions 1 and 2, Theorem 1 implies that the VF $V^\phi$ can be uniquely determined by the immediate cost function $R^\phi$ for a policy $\phi$ if the VF is in an RKHS of a particular class. In fact, the relation between the VF and the immediate cost function in (4) is based on the assumption that the VF is twice continuously differentiable over $\text{int}(\mathcal{X})$, and the verification theorem (cf. [10]) states that, when the immediate cost function and a twice continuously differentiable function satisfying the relation (4) are given under certain conditions, the twice continuously differentiable function is indeed the VF. In Theorem 1, on the other hand, we first restrict the class of the VF by identifying

an RKHS $\mathcal{H}_V$, and then approximate the immediate cost function in the RKHS $\mathcal{H}_R$ any element of which satisfies the relation (4). Because the immediate cost $R^\phi(x(t))$ is observable, we can employ any kernel-based supervised learning to estimate the function $R^\phi$ in the RKHS $\mathcal{H}_R$, such as GPs and KFs, as elaborated later in Section 6.

In the RKHS $\mathcal{H}_R$, an estimate of $R^\phi$ at time instant $n \in \mathbb{Z}_{\geq 0}$ is given by $\hat{R}_n^\phi(x) = \sum_i^r c_i \kappa(x, x_i)$, $c_i \in \mathbb{R}$, $r \in \mathbb{Z}_{\geq 0}$, where $\{x_i\}_{i \in \{1,2,\ldots,r\}} \subset \mathcal{X}$ is the set of samples, and the reproducing kernel $\kappa$ is defined in (9). An estimate of the VF $V^\phi$ at time instant $n \in \mathbb{Z}_{\geq 0}$ is thus immediately obtained by $\hat{V}_n^\phi(x) = U^{-1}(\hat{R}_n^\phi)(x) = \sum_{i=1}^r c_i K(x, x_i)$, where $K$ is defined in (10).

Note, when the system dynamics is described by an ordinary differential equation (i.e., $\eta = 0$), the assumptions that $V^\phi$ is twice continuously differentiable and that $\kappa^V(\cdot, \cdot) \in C^{2 \times 2}(\mathcal{X} \times \mathcal{X})$ are relaxed to that $V^\phi$ is continuously differentiable and that $\kappa^V(\cdot, \cdot) \in C^{2 \times 1}(\mathcal{X} \times \mathcal{X})$, respectively.

As an illustrative example of Theorem 1, we show the case of the linear-quadratic regulator (LQR) below.

**Special case: linear-quadratic regulator**  Consider a *linear* feedback $\phi_{\mathrm{LQR}}$, i.e., $\phi_{\mathrm{LQR}}(x) = -F_{\mathrm{LQR}}x$, $F_{\mathrm{LQR}} \in \mathbb{R}^{n_u \times n_x}$, and a linear system $\dot{x} := \frac{dx}{dt} = A_{\mathrm{LQR}}x + B_{\mathrm{LQR}}u$, where $A_{\mathrm{LQR}} \in \mathbb{R}^{n_x \times n_x}$ and $B_{\mathrm{LQR}} \in \mathbb{R}^{n_x \times n_u}$ are matrices. In this case, we know that the value function $V^{\phi_{\mathrm{LQR}}}$ becomes quadratic with respect to the state variable (cf. [53]). Therefore, we employ an RKHS with a quadratic kernel for $\mathcal{H}_V$, i.e., $\kappa^V(x, y) = (x^\mathsf{T} y)^2$. If we assume that the input space $\mathcal{X}$ is so *large* that the set $\mathrm{span}\{A_{\mathrm{sym}} | A_{\mathrm{sym}} = xx^\mathsf{T}, \exists x \in \mathcal{X}\}$ accommodates any real symmetric matrix, we obtain $\mathcal{H}_V = \{\mathcal{X} \ni x \mapsto x^\mathsf{T} A_{\mathrm{sym}} x | A_{\mathrm{sym}} \text{ is symmetric}\}$.

Moreover, it follows from the product rule of the directional derivative [6] that $K(x, y) = -x^\mathsf{T} \overline{A}_{\mathrm{LQR}} y x^\mathsf{T} y - x^\mathsf{T} y x^\mathsf{T} \overline{A}_{\mathrm{LQR}} y = x^\mathsf{T}(-\overline{A}_{\mathrm{LQR}} yy^\mathsf{T} - yy^\mathsf{T} \overline{A}_{\mathrm{LQR}}^\mathsf{T})x$, where $\overline{A}_{\mathrm{LQR}} := A_{\mathrm{LQR}} - B_{\mathrm{LQR}}F_{\mathrm{LQR}}$. Note $A_{\mathrm{value}}(y) := -\overline{A}_{\mathrm{LQR}} yy^\mathsf{T} - yy^\mathsf{T} \overline{A}_{\mathrm{LQR}}^\mathsf{T}$ is symmetric, implying $K(\cdot, y) \in \mathcal{H}_V$, and we obtain $\kappa(x, y) = -x^\mathsf{T}(\overline{A}_{\mathrm{LQR}}^\mathsf{T} A_{\mathrm{value}}(y) + A_{\mathrm{value}}(y)\overline{A}_{\mathrm{LQR}})x$. Because $A_{\mathrm{cost}}(y) := -\overline{A}_{\mathrm{LQR}}^\mathsf{T} A_{\mathrm{value}}(y) - A_{\mathrm{value}}(y)\overline{A}_{\mathrm{LQR}}$ is symmetric, it follows that $\kappa(\cdot, y) \in \mathcal{H}_V$. If $\overline{A}_{\mathrm{LQR}}$ is stable (Hurwitz), from Theorem 1, the one-to-one correspondence between $\mathcal{H}_V$ and $\mathcal{H}_R$ thus implies that $\mathcal{H}_V = \mathcal{H}_R$. Therefore, we can fully approximate the immediate cost function $R^{\phi_{\mathrm{LQR}}}$ in $\mathcal{H}_R$ if $R^{\phi_{\mathrm{LQR}}}$ is quadratic with respect to the state variable.

Suppose that the immediate cost function is given by $R^{\phi_{\mathrm{LQR}}}(x) = \sum_{i=1}^r c_i \kappa(x, x_i) = x^\mathsf{T} A_{\mathrm{cost}}x$. Then, the estimated value function will be given by $V^{\phi_{\mathrm{LQR}}}(x) = U^{-1}(R^{\phi_{\mathrm{LQR}}})(x) = \sum_{i=1}^r c_i K(x, x_i) = -x^\mathsf{T} \overline{A}_{\mathrm{value}}x$, where $\overline{A}_{\mathrm{LQR}}^\mathsf{T} \overline{A}_{\mathrm{value}} + \overline{A}_{\mathrm{value}}\overline{A}_{\mathrm{LQR}} + \overline{A}_{\mathrm{cost}} = 0$, which is indeed the well-known Lyapunov equation [53]. Unlike Gaussian-kernel cases, we only require a *finite* number of parameters to fully approximate the immediate cost function, and hence is analytically obtainable.

**Barrier-certified policy updates under CT formulation**  Next, we show that the CT formulation makes barrier-certified policy updates computationally more efficient under certain conditions. Assume that the system dynamics is affine in the control, i.e., $h(x, u) = f(x) + g(x)u$, and $\eta = 0$, where $f : \mathbb{R}^{n_x} \to \mathbb{R}^{n_x}$ and $g : \mathbb{R}^{n_x} \to \mathbb{R}^{n_x \times n_u}$, and that the immediate cost $R(x, u)$ is given by $Q(x) + \frac{1}{2}u^\mathsf{T} Mu$, where $Q : \mathbb{R}^{n_x} \to \mathbb{R}$, and $M \in \mathbb{R}^{n_u \times n_u}$ is a positive definite matrix. Then, any Lipschitz continuous policy $\phi : \mathcal{X} \to \mathcal{U}$ satisfying $\phi(x) \in \mathcal{S}(x) := \left\{u \in \mathcal{U} \mid \frac{\partial b(x)}{\partial x}f(x) + \frac{\partial b(x)}{\partial x}g(x)u + \alpha(b(x)) \geq 0\right\}$ renders the set $\mathcal{C}$ forward invariant [50], i.e., the state trajectory remains inside the set $\mathcal{C}$, where $\alpha : \mathbb{R} \to \mathbb{R}$ is strictly increasing and $\alpha(0) = 0$. Taking this constraint into account, the barrier-certified greedy policy update is given by

$$\phi^+(x) = \underset{u \in \mathcal{S}(x)}{\mathrm{argmin}} \left[\frac{1}{2}u^\mathsf{T} Mu + \frac{\partial V^\phi(x)}{\partial x}g(x)u\right], \tag{11}$$

which is, by virtue of the CT formulation, a quadratic programming (QP) problem at $x$ when $\mathcal{U} \subset \mathbb{R}^{n_u}$ defines affine constraints (see Appendix C in the supplementary document). The space of allowable policies is thus given by $\Gamma := \{\phi \in \Phi \mid \phi(x) \in \mathcal{S}(x), \forall x \in \mathcal{X}\}$. When $\eta \neq 0$ and the dynamics is learned by GPs as in [30], the work in [47] provides a barrier certificate for uncertain dynamics. Note, one can also employ a function approximator or add noises to the greedily updated policy to avoid

unstable performance and promote exploration (see e.g., [8]). To see if the updated policy $\phi^+$ remains in the space of Lipschitz continuous policies $\Phi$, i.e., $\Gamma \subset \Phi$, we present the following proposition.

**Proposition 1.** *Assume the conditions in Theorem 1. Assume also that $\mathcal{U} \subset \mathbb{R}^{n_u}$ defines affine constraints, and that $f$, $g$, $\alpha$, and the derivative of $b$ are Lipschitz continuous over $\mathcal{X}$. Then, the policy $\phi^+$ defined in (11) is Lipschitz continuous over $\mathcal{X}$ if the* width of a feasible set[3] *is strictly larger than zero over $\mathcal{X}$.*

*Proof.* See Appendix D in the supplementary document. □

Note, if $\mathcal{U} \subset \mathbb{R}^{n_x}$ defines the bounds of each entry of control inputs, it defines affine constraints. Lastly, the width of a feasible set is strictly larger than zero if $\mathcal{U}$ is sufficiently large and $\frac{\partial b(x)}{\partial x} g(x) \neq 0$.

We will further clarify the relations of the proposed theoretical framework to existing works below.

## 5  Relations to existing works

First, our proposed framework takes advantage of the capability of learning complicated functions and nonparametric flexibility of RKHSs, and reproduces some of the existing *model-based* DT-VF approximation techniques (see Appendix E in the supplementary document). Note that some of the existing DT-VF approximations in RKHSs, such as GPTD [9], also work for model-free cases (see [27] for model-free adaptive DT action-value function approximation, for example). Second, since the RKHS $\mathcal{H}_R$ for learning is explicitly defined in our framework, any kernel-based method and its convergence/tracking analyses are directly applicable. While, for example, the work in [17], which aims at attaining a sparse representation of the unknown function in an online fashion in RKHSs, was extended to the policy evaluation [18] by addressing the double-sampling problem, our framework does not suffer from the double-sampling problem, and hence any kernel-based online learning (e.g., [17, 51, 39]) can be straightforwardly applied. Third, when the time interval is small, DT formulations become susceptible to errors, while CT formulations are immune to the choice of the time interval. Note, on the other hand, a larger time interval poorly represents the system dynamics evolving in continuous time. Lastly, barrier certificates are efficiently incorporated in our CT framework through QPs under certain conditions, and state constraints are implicitly taken into account. Stochastic optimal control such as the work in [43, 42] requires sample trajectories over predefined finite time horizons and the value is computed along the trajectories while the VF is estimated in an RKHS even without having to follow the trajectory in our framework.

## 6  Applications and practical implementation

We apply the theory presented in the previous section to the Gaussian kernel case and introduce CTGP as an example, and present a practical implementation. Assume that $A(x, u) \in \mathbb{R}^{n_x \times n_x}$ is diagonal, for simplicity. The Gaussian kernel is given by $\kappa^V(x, y) := \frac{1}{(2\pi\sigma^2)^{L/2}} \exp\left(-\frac{\|x - y\|^2_{\mathbb{R}^{n_x}}}{2\sigma^2}\right)$, $x, y \in \mathcal{X}$, $\sigma > 0$. Given Gaussian kernel $\kappa^V(x, y)$, the reproducing kernel $\kappa(x, y)$ defined in (9) is derived as $\kappa(x, y) = a(x, y)\kappa^V(x, y)$, where $a : \mathcal{X} \times \mathcal{X} \to \mathbb{R}$ is a real-valued function (see Appendix F in the supplementary document for the explicit form of $a(x, y)$).

**CTGP**  One of the celebrated properties of GPs is their Bayesian formulation, which enables us to deal with uncertainty through credible intervals. Suppose that the observation $d$ at time instant $n \in \mathbb{Z}_{\geq 0}$ contains some noise $\epsilon \in \mathbb{R}$, i.e., $d(x) = R^\phi(x) + \epsilon$, $\epsilon \sim \mathcal{N}(0, \mu_o^2)$, $\mu_o \geq 0$. Given data $d_N := [d(x_0), d(x_1), \ldots, d(x_N)]^\mathsf{T}$ for some $N \in \mathbb{Z}_{\geq 0}$, we can employ GP regression to obtain the mean $m(x_*)$ and the variance $\mu^2(x_*)$ of $\hat{R}^\phi(x_*)$ at a point $x_* \in \mathcal{X}$ as

$$m(x_*) = k_*^\mathsf{T}(G + \mu_o^2 I)^{-1} d_N, \quad \mu^2(x_*) = \kappa(x_*, x_*) - k_*^\mathsf{T}(G + \mu_o^2 I)^{-1} k_*, \qquad (12)$$

where $I$ is the identity matrix, $k_* := [\kappa(x_*, x_0), \kappa(x_*, x_1), \ldots, \kappa(x_*, x_N)]^\mathsf{T}$, and the $(i, j)$ entry of $G \in \mathbb{R}^{(N+1) \times (N+1)}$ is $\kappa(x_{i-1}, x_{j-1})$. Then, by the existence of the inverse operator $U^{-1}$, the mean $m^V(x_*)$ and the variance $\mu^{V2}(x_*)$ of $\hat{V}^\phi(x_*)$ at a point $x_* \in \mathcal{X}$ can be given by

$$m^V(x_*) = K_*^{V\mathsf{T}}(G + \mu_o^2 I)^{-1} d_N, \ \mu^{V2}(x_*) = \kappa^V(x_*, x_*) - K_*^{V\mathsf{T}}(G + \mu_o^2 I)^{-1} K_*^V, \quad (13)$$

Table 2: Comparisons of the cumulative costs and numbers of times the observed velocities became lower than $-0.05$ with and without barrier certificates

|  | CTKF | GPTD_1 | DTKF_1 | CTGP | GPTD_20 | DTKF_20 |
|---|---|---|---|---|---|---|
| Cumulative cost | 114.2 | 299.1 | 299.1 | 82.2 | 89.2 | 90.4 |
| With barrier | 0 (times) | 0 (times) | 0 (times) | 0 (times) | 0 (times) | 0 (times) |
| Without barrier | 0 (times) | 0 (times) | 0 (times) | 10 (times) | 20 (times) | 20 (times) |

where $K_*^V := [K(x_*, x_0), K(x_*, x_1), \ldots, K(x_*, x_N)]^\mathsf{T}$ (see Appendix G in the supplementary document for more details).

# 7 Numerical Experiment

In this section, we first show the validity of the proposed CT framework and its advantage over DT counterparts when the time interval is small, and then compare CTGP and GPTD for RL on a simulated inverted pendulum. In both of the experiments, the coherence-based sparsification [33] in the RKHS $\mathcal{H}_R$ is employed to curb the growth of the dictionary size.

**Policy evaluations: comparison of CT and DT approaches** We show that our CT approaches are advantageous over DT counterparts in terms of susceptibility to errors, by using MountainCarContinuous-v0 in OpenAI Gym [7] as the environment. The state is given by $x(t) := [\mathrm{x}(t), \mathrm{v}(t)]^\mathsf{T} \in \mathbb{R}^2$, where $\mathrm{x}(t)$ and $\mathrm{v}(t)$ are the position and the velocity of the car, and the dynamics is given by $dx = \begin{bmatrix} \mathrm{v}(t) \\ -0.0025 \cos{(3\mathrm{x}(t))} \end{bmatrix} dt + \begin{bmatrix} 0 \\ 0.0015 \end{bmatrix} u(t)dt$, where $u(t) \in [-1.0, 1.0]$. The position and the velocity are clipped to $[-0.07, 0.07]$ and $[-1.2, 0.6]$, respectively, and the goal is to reach the position $\mathrm{x} = 0.45$. In the simulation, the control cycle (i.e., the frequency of applying control inputs and observing the states and costs) is set to $1.0$ second. The observed immediate cost is given by $R(x(t), u(t)) + \epsilon = 1 + 0.001u^2(t) + \epsilon$ for $\mathrm{x}(t) < 0.45$ and $R(x(t), u(t)) + \epsilon = 0.001u^2(t) + \epsilon$ for $\mathrm{x}(t) \geq 0.45$, where $\epsilon \sim \mathcal{N}(0, 0.1^2)$. Note the immediate cost for the DT cases is given by $(R(x(t), u(t)) + \epsilon)\Delta t$, where $\Delta t$ is the time interval. For policy evaluations, we use the policy obtained by RL based on the cross-entropy methods[4], and the four methods, CTGP, KF-based CT-VF approximation (CTKF), GPTD, and KF-based DT-VF approximation (DTKF), are used to learn value functions associated with the policy by executing the current policy for five episodes, each of which terminates whenever $t = 300$ or $\mathrm{x}(t) \geq 0.45$. GP-based techniques are expected to handle the random component $\epsilon$ added to the immediate cost. The new policies are then obtained by the barrier-certified policy updates under CT formulations, and these policies are evaluated for five times. Here, the barrier function is given by $b(x) = 0.05 + \mathrm{v}$, which prevents the velocity from becoming lower than $-0.05$. Figure 3 compares the value functions[5] learned by each method for the time intervals $\Delta t = 20.0$ and $\Delta t = 1.0$. We observe that the DT approaches are sensitive to the choice of $\Delta t$. Table 2 compares the cumulative costs averaged over five episodes for each method and for different time intervals and the numbers of times we observed the velocity being lower than $-0.05$ when the barrier certificate is employed and unemployed. (Numbers associated with the algorithm names indicate the lengths of the time intervals.) Note that the cumulative costs are calculated by summing up the immediate costs multiplied by the duration of each control cycle, i.e., we discretized the immediate cost based on the control cycle. It is observed that the CT approach is immune to the choice of $\Delta t$ while the performance of the DT approach degrades when the time interval becomes small, and that the barrier-certified policy updates work efficiently.

**Reinforcement learning: inverted pendulum** We show the advantage of CTGP over GPTD when the time interval for the estimation is small. Let the state $x(t) := [\theta(t), \omega(t)]^\mathsf{T} \in \mathbb{R}^2$ consists of the angle $\theta(t)$ and the angular velocity $\omega(t)$ of an inverted pendulum, and we consider the dynamics: $dx = \begin{bmatrix} \omega(t) \\ \frac{g}{\ell}\sin(\theta(t)) - \frac{\rho}{m\ell^2}\omega(t) \end{bmatrix} dt + \begin{bmatrix} 0 \\ \frac{1}{m\ell^2} \end{bmatrix} u(t)dt + 0.01 I dw$, where $g = 9.8$, $m = 1$, $\ell = 1$, $\rho = 0.01$. The Brownian motion may come from outer disturbances and/or modeling error. In the simulation, the time interval $\Delta t$ is set to $0.01$ seconds, and the simulated dynamics evolves

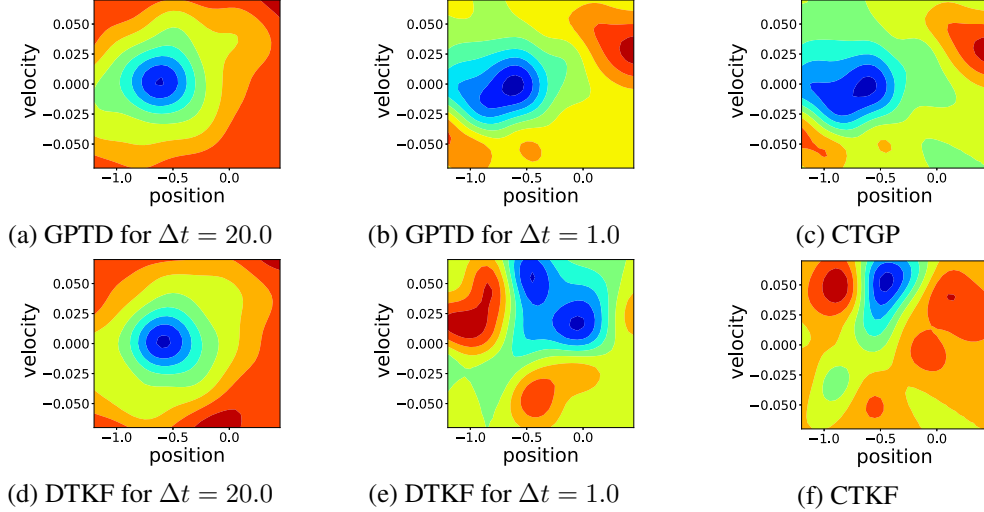

(a) GPTD for $\Delta t = 20.0$      (b) GPTD for $\Delta t = 1.0$      (c) CTGP

(d) DTKF for $\Delta t = 20.0$      (e) DTKF for $\Delta t = 1.0$      (f) CTKF

Figure 3: Illustrations of the value functions obtained by CTGP, CTKF, GPTD, and DTKF for time intervals $\Delta t = 20.0$ and $\Delta t = 1.0$. The policy is obtained by RL based on the cross-entropy method. CT approaches are not affected by the choice of $\Delta t$.

by $\Delta x = h(x(t), u(t))\Delta t + \sqrt{\Delta t}\eta(x(t), u(t))\epsilon_w$, where $\epsilon_w \sim \mathcal{N}(0, I)$. In this experiment, the task is to stabilize the inverted pendulum at $\theta = 0$. The observed immediate cost is given by $R(x(t), u(t)) + \epsilon = 1/(1 + e^{-10(\theta(t)-\pi/16)}) + 100/(1 + e^{-10(\theta(t)-\pi/6)}) + 0.05u^2(t) + \epsilon$, where $\epsilon \sim \mathcal{N}(0, 0.1^2)$. A trajectory associated with the current policy is generated to learn the VF. The trajectory terminates when $|\theta(t)| > \pi/4$ and restarts from a random initial angle. After 10 seconds, the policy is updated. To evaluate the current policy, average time over five episodes in which the pendulum stays up ($|\theta(t)| \leq \pi/4$) when initialized as $\theta(0) \in [-\pi/6, \pi/6]$ is used. Figure 4 compares this average time of CTGP and GPTD up to five updates with standard deviations until when stable policy improvement becomes difficult without some heuristic techniques such as adding noises to policies. Note that the same seed for the random number generator is used for the initializations of both of the two approaches. It is observed that GPTD fails to improve policies. The large credible interval of CTGP is due to the random initialization of the state.

## 8 Conclusion and future work

We presented a novel theoretical framework that renders the CT-VF approximation problem simultaneously solvable in an RKHS by conducting kernel-based supervised learning for the immediate cost function in the properly defined RKHS. Our CT framework is compatible with rich theories of control, including control barrier certificates for safety-critical applications. The validity of the proposed framework and its advantage over DT counterparts when the time interval is small were verified experimentally on the classical Mountain Car problem and a simulated inverted pendulum.

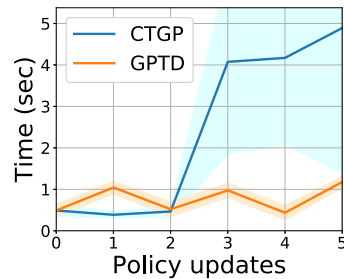

Figure 4: Comparison of time up to which the pendulum stays up between CTGP and GPTD for the inverted pendulum ($\pm$ std. deviation).

There are several possible directions to explore as future works; First, we can employ the state-of-the-art kernel methods within our theoretical framework or use other variants of RL, such as actor-critic methods, to improve practical performances. Second, we can consider uncertainties in value function approximation by virtue of the RKHS-based formulation, which might be used for safety verifications. Lastly, it is worth further explorations of advantages of CT formulations for physical tasks.

**Acknowledgments**

This work was partially conducted when M. Ohnishi was at the GRITS Lab, Georgia Institute of Technology; M. Ohnishi thanks the members of the GRITS Lab, including Dr. Li Wang, and Prof. Magnus Egerstedt for discussions regarding barrier functions. M. Yukawa was supported in part by KAKENHI 18H01446 and 15H02757, M. Johansson was supported in part by the Swedish Research Council and by the Knut and Allice Wallenberg Foundation, and M. Sugiyama was supported in part by KAKENHI 17H00757. Lastly, the authors thank all of the anonymous reviewers for their very insightful comments.

## Footnotes

[1]The cost function might be obtained by the negation of the reward function.

[2] See, for example, [10, Chapter IV],[19], for more detailed arguments about the conditions under which twice continuous differentiability is guaranteed.

[3]*Width* indicates how much control margin is left for the strictest constraint, as defined in [24, Equation 21].

[4]We used the code in https://github.com/udacity/deep-reinforcement-learning/blob/master/cross-entropy/CEM.ipynb offered by Udacity. The code is based on PyTorch [31].

[5]We used "jet colormaps" in Python Matplotlib for illustrating the value functions.

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
