[Supplementary Material · NeurIPS2018_supplementary_camera_ready.pdf]

# Continuous-time Value Function Approximation in Reproducing Kernel Hilbert Spaces –Supplementary material–

## A    Tools to prove Theorem 1

Some known properties of RKHSs and Dynkin's formula which will be used to prove Theorem 1 are given below.

**Lemma A.1** ([6, Theorem 2]). *Let $\mathcal{X} \subset \mathbb{R}^{n_x}$ be any set with nonempty interior. Then, the RKHS associated with the Gaussian kernel for an arbitrary scale parameter $\sigma > 0$ does not contain any polynomial on $\mathcal{X}$, including the nonzero constant function.*

**Proposition A.1** ([13, Theorem 1]). *Let $(\mathcal{H}, \langle \cdot, \cdot \rangle_{\mathcal{H}})$ be the RKHS associated with a Mercer kernel $\kappa \in C^{2s}(\mathcal{X} \times \mathcal{X})$, $s \in \mathbb{Z}_{\geq 0}$, where $\mathcal{X} \subset \mathbb{R}^{n_x}$ is compact with nonempty interior. Then, $(D^\alpha \kappa)_x \in \mathcal{H}$, $\forall x \in \mathcal{X}$, $\alpha \in I_s$, and*

$$D^\alpha \varphi(x) = \langle (D^\alpha \kappa)_x, \varphi \rangle_{\mathcal{H}}, \ \forall x \in \mathcal{X}, \ \varphi \in \mathcal{H}. \tag{A.1}$$

**Dynkin's formula**    Under Assumption 1, we obtain Dynkin's formula (cf. [9, Theorem 7.3.3, Theorem 7.4.1]):

$$E_x\left[\Psi(x(t_1))\right] - \Psi(x) = -E_x\left[\int_0^{t_1} \mathcal{G}(\Psi)(x(q))dq\right], \ \ \forall t_1 \in [0, \infty), \tag{A.2}$$

for any $x \in \mathbb{R}^{n_x}$ and for any $\Psi \in C_0^2(\mathbb{R}^{n_x})$, i.e., $\Psi \in C^2(\mathbb{R}^{n_x})$ and $\Psi$ has compact support, where $\mathcal{G}$ is defined in (5). Moreover, it holds [3, Chapter III.3], for $\beta > 0$, that

$$e^{-\beta t_1}E_x\left[\Psi(x(t_1))\right] - \Psi(x) = -E_x\left[\int_0^{t_1} e^{-\beta q}\left[\beta I_{\mathrm{op}} + \mathcal{G}\right](\Psi)(x(q))dq\right], \ \ \forall t_1 \in [0, \infty). \tag{A.3}$$

When $t_1$ is the first exit time of a bounded set[1], then the condition for $\Psi$ is weakened into $\Psi \in C^2$ over the bounded set (see the remark of [9, Theorem 7.4.1]). See Figure B.1 for an intuition of the expectation $E_x$ taken for the trajectories of the state starting from $x$.

## B    Proof of Theorem 1

Note first that the operator $U$ is well defined because $D^\alpha \varphi^V$ exists for any $\varphi^V \in \mathcal{H}_V$ from Proposition A.1. We show that $U$ is bijective linear, and then show that the reproducing kernel in $\mathcal{H}_R$ is given by (9).

Figure B.1: Given $x \in \mathbb{R}^{n_x}$, the state evolves from $x(0) = x$ by following the SDE, and $x(\Delta t)$ is thus stochastic. The expectation $E_x$ is taken for all the trajectories.

**Proof of (a)** Because the operator $U$ is surjective by definition of $\mathcal{H}_R$, we show that $U$ is injective. The operator $U$ is linear because the operator $D^\alpha$ is linear by (A.1) in Proposition A.1:

$$
\begin{aligned}
D^\alpha(\nu_1\varphi_1^V + \nu_2\varphi_2^V)(x) &= \left\langle (D^\alpha\kappa)_x, \nu_1\varphi_1^V + \nu_2\varphi_2^V \right\rangle_{\mathcal{H}_V} \\
&= \nu_1 \left\langle (D^\alpha\kappa)_x, \varphi_1^V \right\rangle_{\mathcal{H}_V} + \nu_2 \left\langle (D^\alpha\kappa)_x, \varphi_2^V \right\rangle_{\mathcal{H}_V} \\
&= \left[ \nu_1 D^\alpha\varphi_1^V + \nu_2 D^\alpha\varphi_2^V \right](x), \ \forall x \in \mathcal{X}, \ \forall \nu_i \in \mathbb{R}, \ \forall \varphi_i \in \mathcal{H}_V, \ i \in \{1,2\}.
\end{aligned}
\tag{B.1}
$$

Hence, it is sufficient to show that $\ker(U) = 0$ [11]. Suppose that $U(\varphi^V)(x) = 0, \ \forall x \in \mathcal{X}$. It follows that $[\beta I_{\mathrm{op}} + \mathcal{G}](\varphi^V)(x) = 0, \ \forall x \in \mathrm{int}(\mathcal{X})$, where $\mathcal{G}$ is defined in (6). Under Assumptions 1 and 2 (i.e., compactness of $\mathcal{X}$), Dynkin's fomula (A.2) and (A.3) can be applied to $\varphi^V$ over the bounded set $\mathrm{int}(\mathcal{X})$ because $\varphi^V|_{\mathrm{int}(\mathcal{X})} \in C^2(\mathrm{int}(\mathcal{X}))$. (i) When the discount factor $\beta > 0$, we apply (A.3) to $\varphi^V$. Under Assumption 1, we can consider the time being taken to infinity. Because $[\beta I_{\mathrm{op}} + \mathcal{G}](\varphi^V)(x) = 0, \ \forall x \in \mathrm{int}(\mathcal{X})$, we obtain

$$
\begin{aligned}
\varphi^V(x) &= \lim_{t_1 \to \infty} E_x \left[ \int_0^{t_1} e^{-\beta q} \cdot 0 \, dq \right] + \lim_{t_1 \to \infty} e^{-\beta t_1} E_x \left[ \varphi^V(x(t_1)) \right] \\
&= \lim_{t_1 \to \infty} e^{-\beta t_1} E_x \left[ \varphi^V(x(t_1)) \right], \ \forall x \in \mathrm{int}(\mathcal{X}).
\end{aligned}
\tag{B.2}
$$

Under Assumption 2 (i.e., compactness of $\mathcal{X}$ and invariance of $\mathrm{int}(\mathcal{X})$), $\lim_{t_1 \to \infty} E_x \left[ \varphi^V(x(t_1)) \right]$ is bounded, from which it follows that $\varphi^V(x) = 0$ over $\mathrm{int}(\mathcal{X})$. (ii) When $\beta = 0$ and $x_{t \to \infty}$ is stochastically asymptotically stable over $\mathrm{int}(\mathcal{X})$, we apply (A.2) to $\varphi^V$. Because $\mathcal{G}(\varphi^V)(x) = 0, \ \forall x \in \mathrm{int}(\mathcal{X})$, we obtain

$$
\varphi^V(x) = \lim_{t_1 \to \infty} E_x \left[ \varphi^V(x(t_1)) \right] = \varphi^V(x_{t \to \infty}), \ \forall x \in \mathrm{int}(\mathcal{X}),
\tag{B.3}
$$

which implies that $\varphi^V$ is constant over $\mathrm{int}(\mathcal{X})$. From Lemma A.1, however, it follows that $\varphi^V(x) = 0, \ \forall x \in \mathrm{int}(\mathcal{X})$, when $\mathcal{H}_V$ is a Gaussian RKHS. Therefore, continuity of an element of $\mathcal{H}_V$ implies that $\varphi^V(x) = 0$ over $\mathcal{X}$ for both cases (i) and (ii), which verifies $\ker(U) = 0$. Thus, the correspondence between $\varphi^V \in \mathcal{H}_V$ and $\varphi = U(\varphi^V) \in \mathcal{H}_R$ is one-to-one, and inner product preserves by definition (8).

**Proof of (b)** We show that $\mathcal{H}_R$ is an RKHS. Under Assumptions 1 and 2 (i.e., compactness of $\mathcal{X}$), $h^\phi$ and $A^\phi$ are continuous and hence are bounded over $\mathcal{X}$. Thus, Proposition A.1 (i.e., $(D^\alpha\kappa)_x \in \mathcal{H}_V$) implies that $K(\cdot, x) \in \mathcal{H}_V$. Therefore, it follows that $\kappa(\cdot, x) \in \mathcal{H}_R$. Moreover, from (A.1) in Proposition A.1, we obtain

$$
U(\varphi^V)(x) = \left\langle \varphi^V, K(\cdot, x) \right\rangle_{\mathcal{H}_V},
\tag{B.4}
$$

from which we obtain

$$
\langle \kappa(\cdot, x), \kappa(\cdot, y) \rangle_{\mathcal{H}_R} = \langle K(\cdot, x), K(\cdot, y) \rangle_{\mathcal{H}_V} = U(K(\cdot, y))(x) = \kappa(x, y),
\tag{B.5}
$$

and that

$$
\langle \varphi, \kappa(\cdot, x) \rangle_{\mathcal{H}_R} = \left\langle U^{-1}(\varphi), K(\cdot, x) \right\rangle_{\mathcal{H}_V} = U(U^{-1}(\varphi))(x) = \varphi(x), \ \forall \varphi \in \mathcal{H}_R, \forall x \in \mathcal{X}.
\tag{B.6}
$$

Therefore, $\kappa(\cdot, \cdot) : \mathcal{X} \times \mathcal{X} \to \mathbb{R}$ is the reproducing kernel with which the RKHS $\mathcal{H}_R$ is associated.

## C  Barrier-certified policy update

As illustrated in Figure 2, the space of the allowable policies is given by $\Gamma$ instead of $\Phi$ to implicitly enforce state constraints. Therefore, the greedy policy update is conducted assuming that $\Gamma$ is the whole policy space. We assume that $\eta$ and hence $A$ is independent on $u$, $h(x, u) = f(x) + g(x)u$, and the immediate cost $R(x, u)$ is given by $Q(x) + \frac{1}{2}u^\mathsf{T} Mu$ ($\eta$ was assumed to be 0 in the main text for simplicity of barrier certificates). Given the current policy $\phi$, from (4), the greedy policy update at state $x \in \text{int}(X)$ is given by

$$\phi^+(x) = \operatorname*{argmin}_{u \in \mathcal{S}(x)} \left[ \frac{1}{2}\text{tr}\left[ \frac{\partial^2 V^\phi(x)}{\partial x^2} A(x, u) \right] + \frac{\partial V^\phi(x)}{\partial x} h(x, u) + R(x, u) \right], \tag{C.1}$$

$$= \operatorname*{argmin}_{u \in \mathcal{S}(x)} \left[ \frac{1}{2}u^\mathsf{T} Mu + \frac{\partial V^\phi(x)}{\partial x} g(x)u \right]. \tag{C.2}$$

The simplicity of this optimization problem comes from the CT formulation of the VF and barrier certificates.

## D  Lipschitz continuity of barrier-certified policies

Proposition 1 is based on the following theorem.

**Theorem D.1** ([7, Theorem 1]). *Consider the QP:*

$$u^*(x) = \operatorname*{argmin}_{u \in \mathbb{R}^{n_u}} u^\mathsf{T} H(x)u + 2v(x)^\mathsf{T} u \tag{D.1}$$

$$\text{s.t.}\ \ A_{(\leq)}(x)u \leq a_{(\leq)}(x)$$
$$A_{(=)}(x)u = a_{(=)}(x),$$

*where $H$, $v$, $A_{(\leq)}$, $A_{(=)}$, $a_{(\leq)}$, and $a_{(=)}$ are continuous functions, and define the width of a feasible set as the unique solution to the following linear program:*

$$u_w(x) = \max_{[u^\mathsf{T}, u_w]^\mathsf{T} \in \mathbb{R}^{n_u+1}} u_w \tag{D.2}$$

$$\text{s.t.}\ \ A_{(\leq)}(x)u + [u_w, u_w, \ldots, u_w]^\mathsf{T} \leq a_{(\leq)}(x)$$
$$A_{(=)}(x)u = a_{(=)}(x).$$

*Suppose that the following conditions hold at a point $x^* \in \mathcal{X}$:*

1. *$u_w(x^*) > 0$*

2. *$A_{(=)}(x)$ has full row rank*

3. *$A_{(\leq)}(x)$, $A_{(=)}(x)$, $a_{(\leq)}(x)$, and $a_{(=)}(x)$ are Lipschitz continuous at $x^*$*

4. *$H(x^*) = H^\mathsf{T}(x^*)$ and is positive definite*

5. *$H(x)$ and $v(x)$ are Lipschitz continuous at $x^*$*

*Then the feedback $u^*(x)$ defined in (D.1) is unique and Lipschitz continuous with respect to the state at $x^*$.*

We also use the following facts to prove Proposition 1.

**Fact D.1.** *The product of two Lipschitz continuous functions over a bounded set $\mathcal{X}$ is also Lipschitz continuous over $\mathcal{X}$.*

**Fact D.2.** *Given a compact set $\mathcal{X}$, a function which is Lipschitz continuous at any point $x \in \mathcal{X}$ is Lipschitz continuous over $\mathcal{X}$.*

**Fact D.3.** *Suppose that $b : \mathcal{X} \to \mathbb{R}$ is Lipschitz continuous over a set $\mathcal{X}$ and that $\alpha$ is Lipschitz continuous over $\mathbb{R}$. Then, the composite function $\alpha \circ b$ is Lipschitz continuous over $\mathcal{X}$.*

We now prove Proposition 1. For the barrier-certified policy update (11), inequality constraints represent the affine constraints $\mathcal{U}$ and the barrier certificates, and there are no equality constraints. It is, however, possible to augment $u \in \mathbb{R}^{n_u}$ and consider the QP:

$$u^*_{\text{aug}}(x) = \underset{u_{\text{aug}} \in \mathbb{R}^{n_u+1}}{\text{argmin}} \ u_{\text{aug}}^\mathsf{T} \begin{bmatrix} H(x) & 0 \\ 0 & 1 \end{bmatrix} u_{\text{aug}} + 2[v(x)^\mathsf{T}, 1] u_{\text{aug}} \tag{D.3}$$

$$\text{s.t.} \ \ \left[A_{(\leq)}(x), \mathbf{0}\right] u_{\text{aug}} \leq [a_{(\leq)}^\mathsf{T}(x), 1]^\mathsf{T}$$

$$[0, 0, \ldots, 1] \, u_{\text{aug}} = 0.$$

Therefore, we can ignore the condition that $A_{(=)}$ has full row rank if there are no equality constraints. Because $f$, $g$, $\alpha$, and the derivative of $b$ are Lipschitz continuous over the compact set $\mathcal{X}$, Fact D.1 and Fact D.3 imply that $\frac{\partial b(x)}{\partial x}g(x)$ and $\frac{\partial b(x)}{\partial x}f(x) + \alpha(b(x))$ are Lipschitz continuous over $\mathcal{X}$. Therefore, $A_\leq(x)$ and $a_\leq(x)$ are Lipschitz continuous over $\mathcal{X}$.

Moreover, because the function $V^\phi$ is in the RKHS $\mathcal{H}_V$ associated with the reproducing kernel $\kappa^V(\cdot, \cdot) \in C^{2 \times 2}(\mathcal{X} \times \mathcal{X})$, $\frac{\partial V^\phi(x)}{\partial x}$ is Lipschitz continuous. Therefore, Lipschitz continuity of $g(x)$ and Fact D.1 imply that $\frac{\partial V^\phi(x)}{\partial x}g(x)$ is Lipschitz continuous over the compact set $\mathcal{X}$.

Lastly, $M$ is positive definite and constant over $\mathcal{X}$.

From Theorem D.1 and Fact D.2, the policy $\phi^+(x)$ defined in (11) is Lipschitz continuous over $\mathcal{X}$ if the width of a feasible set $u_w(x)$ is strictly larger than zero at any point in $\mathcal{X}$.

# E  DT case and its relation to the existing approaches

When the proposed framework is applied to model-based DT-VF approximation in RKHSs, it reproduces some of the existing methods. The Bellman equation of a policy $\phi$ for a DT-VF $\tilde{V}^\phi$ is given by

$$\tilde{V}^\phi(x_n) - \gamma \int_\mathcal{X} \tilde{V}^\phi(x_+) p^\phi(x_+|x_n) dx_+ = \tilde{R}^\phi(x_n) := \tilde{R}(x_n, \phi(x_n)), \tag{E.1}$$

where $\gamma \in [0, 1)$ is the DT discount factor, $x_n$ is the state observed at time instant $n \in \mathbb{Z}_{\geq 0}$, $\tilde{R} : \mathbb{R}^{n_x} \times \mathcal{U} \to \mathbb{R}$ is the average immediate cost function at each time instant, and $p^\phi(x_+|x)$ is the probability that, given a policy $\phi$, the successor state is $x_+$ conditioned on the current state $x$.

**Lemma E.1** ([1, page 35 - Corollary 4]). *Let $\mathcal{H}$ be a Hilbert space associated with the reproducing kernel $\kappa(\cdot, \cdot) : \mathcal{X} \times \mathcal{X} \to \mathbb{R}$. If $\mathcal{X} \subset \mathbb{R}^{n_x}$ is compact and $\kappa(\cdot, x)$ is continuous for any given $x \in \mathcal{X}$, then $\mathcal{H}$ is the space of continuous functions.*

**Theorem E.1.** *Suppose that $\mathcal{H}_{\tilde{V}}$ is an RKHS associated with the reproducing kernel $\kappa^{\tilde{V}}(\cdot, \cdot) : \mathcal{X} \times \mathcal{X} \to \mathbb{R}$. Suppose also that $\mathcal{X} \subset \mathbb{R}^{n_x}$ is compact, and that $\kappa^{\tilde{V}}(\cdot, x)$ is continuous for any given $x \in \mathcal{X}$. Define the operator $\tilde{U}$ as $\tilde{U}(\varphi^{\tilde{V}})(x) := \varphi^{\tilde{V}}(x) - \gamma \int_\mathcal{X} \varphi^{\tilde{V}}(x_+) p^\phi(x_+|x) dx_+$ for any $\varphi^{\tilde{V}} \in \mathcal{H}_{\tilde{V}}$ and for any $x \in \mathcal{X}$, where $\gamma \in [0, 1)$. Then, the following statements hold.*
*(a) The space*

$$\mathcal{H}_{\tilde{R}} := \{\varphi \mid \varphi(x) = \tilde{U}(\varphi^{\tilde{V}})(x), \ \exists \varphi^{\tilde{V}} \in \mathcal{H}_{\tilde{V}}, \forall x \in \mathcal{X}\} \tag{E.2}$$

*is an isomorphic Hilbert space of $\mathcal{H}_{\tilde{V}}$ equipped with the inner product defined by*

$$\langle \varphi_1, \varphi_2 \rangle_{\mathcal{H}_{\tilde{R}}} := \left\langle \varphi_1^{\tilde{V}}, \varphi_2^{\tilde{V}} \right\rangle_{\mathcal{H}_{\tilde{V}}}, \ \varphi_i(x) := \tilde{U}(\varphi_i^{\tilde{V}})(x), \ \forall x \in \mathcal{X}, \ i \in \{1, 2\}. \tag{E.3}$$

*(b) The Hilbert space $\mathcal{H}_{\tilde{R}}$ has the reproducing kernel given by*

$$\tilde{\kappa}(x, y) := \tilde{U}(\tilde{K}(\cdot, y))(x), \ x, y \in \mathcal{X}, \tag{E.4}$$

*where*

$$\tilde{K}(\cdot, x) = \kappa^{\tilde{V}}(\cdot, x) - \gamma m_x^\kappa. \tag{E.5}$$

*Here, $m_x^\kappa \in \mathcal{H}_{\tilde{V}}$ is the embedding satisfying*

$$\left\langle m_x^\kappa, \varphi^{\tilde{V}} \right\rangle_{\mathcal{H}_{\tilde{V}}} = \int_\mathcal{X} \varphi^{\tilde{V}}(x_+) p^\phi(x_+|x) dx_+. \tag{E.6}$$

*Proof.* We show that $\tilde{U}$ is bijective linear, and then show that the reproducing kernel in $\mathcal{H}_{\tilde{R}}$ is given by (E.4).

**Proof of (a)** Because the operator $\tilde{U}$ is surjective by definition of $\mathcal{H}_{\tilde{R}}$, we show that $\tilde{U}$ is injective. The expectation operator and hence $\tilde{U}$ is linear. Hence, it is sufficient to show that $\ker(\tilde{U}) = 0$ [11]. Suppose that $\tilde{U}(\varphi^{\tilde{V}})(x) = 0$, $\forall x \in \mathcal{X}$. It then follows that $\varphi^{\tilde{V}}(x) = \gamma \int_{\mathcal{X}} \varphi^{\tilde{V}}(x_+) p^\phi(x_+|x) dx_+$ for all $x \in \mathcal{X}$. We show that $\varphi^{\tilde{V}} = 0$ by contradiction. Since $\mathcal{X}$ is compact and any $\varphi^{\tilde{V}} \in \mathcal{H}_{\tilde{V}}$ is continuous from Lemma E.1, $\varphi^{\tilde{V}}$ attains the maximum/minimum value at some point $x_{\max}/x_{\min}$, respectively. If $\varphi^{\tilde{V}} \neq 0$, then we obtain $\varphi^{\tilde{V}}(x_{\max}) > 0$ or $\varphi^{\tilde{V}}(x_{\min}) < 0$. If $\varphi^{\tilde{V}}(x_{\max}) > 0$, it follows that

$$\varphi^{\tilde{V}}(x_{\max}) = \gamma \int_{\mathcal{X}} \varphi^{\tilde{V}}(x_+) p^\phi(x_+|x_{\max}) dx_+ < \varphi^{\tilde{V}}(x_{\max}), \tag{E.7}$$

for $\gamma \in [0,1)$, which is contradictory. If $\varphi^{\tilde{V}}(x_{\min}) < 0$, it follows that

$$\varphi^{\tilde{V}}(x_{\min}) = \gamma \int_{\mathcal{X}} \varphi^{\tilde{V}}(x_+) p^\phi(x_+|x_{\min}) dx_+ > \varphi^{\tilde{V}}(x_{\min}), \tag{E.8}$$

for $\gamma \in [0,1)$, which is also contradictory. Hence, $\varphi^{\tilde{V}} = 0$, and $\ker(\tilde{U}) = 0$. Therefore, the correspondence between $\varphi^{\tilde{V}} \in \mathcal{H}_{\tilde{R}}$ and $\varphi = \tilde{U}(\varphi^{\tilde{V}})$ is one-to-one, and inner product preserves by definition (E.3).

**Proof of (b)** We show that $\mathcal{H}_{\tilde{R}}$ is an RKHS. Because $m_x^\kappa \in \mathcal{H}_{\tilde{V}}$ and hence $\tilde{K}(\cdot, x) \in \mathcal{H}_{\tilde{V}}$, it follows that $\tilde{\kappa}(\cdot, x) \in \mathcal{H}_{\tilde{R}}$. Moreover, it holds that

$$\tilde{U}(\varphi^{\tilde{V}})(x) = \left\langle \varphi^{\tilde{V}}, \tilde{K}(\cdot, x) \right\rangle_{\mathcal{H}_{\tilde{V}}}, \tag{E.9}$$

from which we obtain

$$\langle \tilde{\kappa}(\cdot, x), \tilde{\kappa}(\cdot, y) \rangle_{\mathcal{H}_{\tilde{R}}} = \left\langle \tilde{K}(\cdot, x), \tilde{K}(\cdot, y) \right\rangle_{\mathcal{H}_{\tilde{V}}} = \tilde{U}(\tilde{K}(\cdot, y))(x) = \tilde{\kappa}(x, y), \tag{E.10}$$

and that

$$\langle \varphi, \tilde{\kappa}(\cdot, x) \rangle_{\mathcal{H}_{\tilde{R}}} = \left\langle \tilde{U}^{-1}(\varphi), \tilde{K}(\cdot, x) \right\rangle_{\mathcal{H}_{\tilde{V}}} = \tilde{U}(\tilde{U}^{-1}(\varphi))(x) = \varphi(x), \ \forall \varphi \in \mathcal{H}_{\tilde{R}}, \forall x \in \mathcal{X}. \tag{E.11}$$

Therefore, $\tilde{\kappa}(\cdot, \cdot) : \mathcal{X} \times \mathcal{X} \to \mathbb{R}$ is the reproducing kernel with which the RKHS $\mathcal{H}_{\tilde{R}}$ is associated. $\quad\square$

**Remark E.1.** *If $x_+$ is deterministically obtained, $m_x^\kappa = \kappa^{\tilde{V}}(\cdot, x_+)$ for some $x_+ \in \mathcal{X}$.*

**Remark E.2.** *Note that the work in [8] considers model-free DT action-value function approximation and defines an RKHS over $\mathcal{Z}^2$, where $\mathcal{Z} := \mathcal{X} \times \mathcal{U}$, while Theorem E.1 is for model-based DT-VF approximation.*

Provided Theorem E.1, we show how some of the existing model-based DT-VF approximation methods can be reproduced.

**Online gradient descent on a sequence of Bellman loss functions** In [12], the residual gradient algorithm is viewed as running online gradient descent on the Bellman loss, and the following update equation is given:

$$\hat{\tilde{V}}_{n+1}^\phi = \hat{\tilde{V}}_n^\phi - \lambda_n e_n^B (\nabla_{\tilde{V}} \hat{\tilde{V}}_n^\phi(x_n) - \gamma \nabla_{\tilde{V}} \hat{\tilde{V}}_n^\phi(x_{n+1})), \tag{E.12}$$

where $e_n^B := \hat{\tilde{V}}_n^\phi(x_n) - \gamma \hat{\tilde{V}}_n^\phi(x_{n+1}) - \tilde{R}^\phi(x_n)$, and $\nabla_{\tilde{V}} \tilde{V}(x)$ is the functional gradient of the evaluation functional $\tilde{V}(x)$ at a function $\tilde{V}$. Note that it is implicitly assumed that $\tilde{V}$ is in an RKHS $\mathcal{H}_{\tilde{V}}$ and hence $\nabla_{\tilde{V}} \tilde{V}(x) = \kappa^{\tilde{V}}(\cdot, x)$. If, on the other hand, we employ $\mathcal{H}_{\tilde{R}}$ defined in Theorem E.1 as the stage of learning, and apply online gradient descent, we obtain the update rule:

$$\hat{\tilde{R}}_{n+1}^\phi = \hat{\tilde{R}}_n^\phi - \lambda_n (\hat{\tilde{R}}_n^\phi(x_n) - \tilde{R}^\phi(x_n)) \tilde{\kappa}(\cdot, x), \tag{E.13}$$

which results in updating $\tilde{V}_n^\phi$ by (E.12) within $\mathcal{H}_{\tilde{V}}$.

**MDPs with RKHS embeddings** A kernelized version of MDPs has been proposed in [4]. Nonparametric nature of kernelized MDPs enables us to handle complicated distributions, high-dimensional data, and continuous states and actions, and convergence can be analyzed in the infinite sample case [4]. The embedding $m_x^\kappa$ might be learned [10] or gives information about the inaccuracy of a nominal model. The model-based VF approximation for MDPs in an RKHS can also be efficiently conducted in our framework.

**Gaussian process temporal difference algorithm** To address a probabilistic nature of MDPs, Bayesian approach is a natural option, and GPTD was proposed [2]. We consider a path $(x_n)_{n=0,1,\ldots,N}$ of the state. In GPTD, the posterior mean and variance of $\hat{\tilde{V}}^\phi$ at a point $x_* \in \mathcal{X}$ are given by

$$m^{\tilde{V}}(x_*) = k_*^{\tilde{V}\mathsf{T}} H^\mathsf{T} (H\tilde{G}_k H^\mathsf{T} + \Sigma)^{-1} \tilde{d}_{N-1}, \tag{E.14}$$

$$\mu^{\tilde{V}2}(x_*) = \kappa^{\tilde{V}}(x_*, x_*) - k_*^{\tilde{V}\mathsf{T}} H^\mathsf{T} (H\tilde{G}_k H^\mathsf{T} + \Sigma)^{-1} H k_*^{\tilde{V}}, \tag{E.15}$$

where $\tilde{d}_{N-1} \sim \mathcal{N}([\tilde{R}^\phi(x_0), \tilde{R}^\phi(x_1), \ldots, \tilde{R}^\phi(x_{N-1})]^\mathsf{T}, \Sigma)$ for some $N \in \mathbb{Z}_{\geq 0}$, $k_*^{\tilde{V}} := [\kappa^{\tilde{V}}(x_*, x_0), \kappa^{\tilde{V}}(x_*, x_1), \ldots, \kappa^{\tilde{V}}(x_*, x_N)]^\mathsf{T}$, the $(i,j)$ entry of $\tilde{G}_k \in \mathbb{R}^{(N+1)\times(N+1)}$ is $\kappa^{\tilde{V}}(x_{i-1}, x_{j-1})$, $\Sigma \in \mathbb{R}^{N\times N}$ is the covariance matrix of $\tilde{d}_{N-1}$, and

$$H := \begin{bmatrix} 1 & -\gamma & 0 & \ldots & 0 \\ 0 & 1 & -\gamma & \ldots & 0 \\ \vdots & & & & \vdots \\ 0 & 0 & \ldots & 1 & -\gamma \end{bmatrix} \in \mathbb{R}^{N\times(N+1)}. \tag{E.16}$$

If, on the other hand, the RKHS $\mathcal{H}_{\tilde{R}}$ defined in Theorem E.1 is employed as the stage of learning, by letting $m_{x_n}^\kappa = \kappa^{\tilde{V}}(\cdot, x_{n+1})$ for $n = 0, 1, \ldots, N-1$, we obtain

$$m^{\tilde{V}}(x_*) = K_*^{\tilde{V}\mathsf{T}} (\tilde{G} + \Sigma)^{-1} \tilde{d}_N, \tag{E.17}$$

$$\mu^{\tilde{V}2}(x_*) = \kappa^{\tilde{V}}(x_*, x_*) - K_*^{\tilde{V}\mathsf{T}} (\tilde{G} + \Sigma)^{-1} K_*^{\tilde{V}}, \tag{E.18}$$

where $K_*^{\tilde{V}} := [\tilde{K}(x_*, x_0), \tilde{K}(x_*, x_1), \ldots, \tilde{K}(x_*, x_{N-1})]^\mathsf{T}$, and the $(i,j)$ entry of $\tilde{G} \in \mathbb{R}^{N\times N}$ is $\tilde{\kappa}(x_{i-1}, x_{j-1})$, which result in the same values as GPTD.

As we have shown, Theorem E.1 unifies model-based DT-VF approximation methods working in RKHSs. As such, it is able to analyze tracking/convergence etc. straightforwardly by applying already established arguments of kernel-based methods. The present study enables us to conduct CT-VF approximation in RKHSs, and can also be viewed as a CT version of this result by utilizing a partial derivative reproducing property of certain classes of RKHSs.

# F  Derivative of a Gaussian kernel

The function $a(x, y)$ appearing in Section 6 is given by

$$a(x, y) = \beta^2 - \beta \sum_{i=1}^{n_x} a_1^i(x, y)\{h^{\phi^i}(y) - h^{\phi^i}(x)\} - \frac{\beta}{2} \sum_{i=1}^{n_x} a_2^i(x, y)\{A_{i,i}^\phi(y) + A_{i,i}^\phi(x)\}$$

$$+ \sum_{i,j=1}^{n_x} a_{1,1}^{j,i}(x, y) h^{\phi^i}(y) h^{\phi^j}(x) + \frac{1}{2} \sum_{i,j=1}^{n_x} a_{1,2}^{j,i}(x, y)\{A_{i,i}^\phi(y) h^{\phi^j}(x) - A_{j,j}^\phi(x) h^{\phi^i}(y)\}$$

$$+ \frac{1}{4} \sum_{i,j=1}^{n_x} a_{2,2}^{j,i}(x, y) A_{i,i}^\phi(y) A_{j,j}^\phi(x), \tag{F.1}$$

where $a_1^i$, $a_2^i$, $a_{1,1}^{j,i}$, $a_{1,2}^{j,i}$, and $a_{2,2}^{j,i}$ are defined as

$$(D^{e_i}\kappa^V)_y(x) = a_1^i(x, y)\kappa^V(x, y) := \frac{-(y^i - x^i)}{\sigma^2} \kappa^V(x, y), \tag{F.2}$$

$$(D^{2e_i}\kappa^V)_y(x) = a_2^i(x,y)\kappa^V(x,y) := \frac{(y^i - x^i)^2 - \sigma^2}{\sigma^4}\kappa^V(x,y), \tag{F.3}$$

$$D^{e_j}(D^{e_i}\kappa^V)_y(x) = \begin{cases} a_{1,1}^{j,i}(x,y)\kappa^V(x,y) := \frac{(y^i-x^i)(x^j-y^j)}{\sigma^4}\kappa^V(x,y), & i \neq j, \\ a_{1,1}^{i,i}(x,y)\kappa^V(x,y) := \frac{(y^i-x^i)(x^i-y^i)+\sigma^2}{\sigma^4}\kappa^V(x,y), & i = j, \end{cases} \tag{F.4}$$

$$D^{e_j}(D^{2e_i}\kappa^V)_y(x) = \begin{cases} a_{1,2}^{j,i}(x,y)\kappa^V(x,y) := \frac{-\{(y^i-x^i)^2-\sigma^2\}(x^j-y^j)}{\sigma^6}\kappa^V(x,y), & i \neq j, \\ a_{1,2}^{i,i}(x,y)\kappa^V(x,y) := \frac{-\{(y^i-x^i)^2-3\sigma^2\}(x^i-y^i)}{\sigma^6}\kappa^V(x,y), & i = j, \end{cases} \tag{F.5}$$

and

$$D^{2e_j}(D^{2e_i}\kappa^V)_y(x) = \begin{cases} a_{2,2}^{j,i}(x,y)\kappa^V(x,y) := \frac{\{(x^i-y^i)^2-\sigma^2\}\{(x^j-y^j)^2-\sigma^2\}}{\sigma^8}\kappa^V(x,y), & i \neq j, \\ a_{2,2}^{i,i}(x,y)\kappa^V(x,y) := \frac{\{(x^i-y^i)^2-6\sigma^2\}(x^i-y^i)^2+3\sigma^4}{\sigma^8}\kappa^V(x,y), & i = j. \end{cases} \tag{F.6}$$

## G Derivations of (13)

The mean $m(x_*)$ and the variance $\mu^2(x_*)$ of $\hat{R}^\phi(x_*)$ at a point $x_* \in \mathcal{X}$ are given as

$$m(x_*) = k_*^{\mathsf{T}}(G + \mu_o^2 I)^{-1}d_N, \tag{G.1}$$

$$\mu^2(x_*) = C_K(x_*, x_*) := \kappa(x_*, x_*) - k_*^{\mathsf{T}}(G + \mu_o^2 I)^{-1}k_*. \tag{G.2}$$

From Appendix B, we know that $U$ is linear and there exists $U^{-1} : \mathcal{H}_R \to \mathcal{H}_V$. Then, by following the arguments in [5, Equation (8)], we obtain

$$m^V(x_*) = U^{-1}(m^V)(x_*) = K_*^{V\mathsf{T}}(G + \mu_o^2 I)^{-1}d_N, \tag{G.3}$$

$$\mu^{V\,2}(x_*) = U^{-1}C_K(x_*, x_*)U^{\mathsf{T}^{-1}} := U^{-1}U_r^{-1}C_K(x_*, x_*), \tag{G.4}$$

where $U_r^{-1}$ acts on the second argument of $C_K(x_*, x_*)$, e.g., $K(x, y) = U_r(\kappa^V(x, \cdot))(y)$. Therefore, we obtain

$$\mu^{V\,2}(x_*) = U^{-1}U_r^{-1}C_K(x_*, x_*) = U_r^{-1}(K(x_*, \cdot))(x_*) - U^{-1}k_*^{\mathsf{T}}(G + \mu_o^2 I)^{-1}k_* U^{\mathsf{T}^{-1}}$$

$$= \kappa^V(x_*, x_*) - K_*^{V\mathsf{T}}(G + \mu_o^2 I)^{-1}K_*^V. \tag{G.5}$$

## H Monotone approximation, strong convergence, and sparsity

In our proposed framework, the immediate cost function is estimated. Therefore, if the monotone approximation property and strong convergence are guaranteed for the immediate cost function in the RKHS $\mathcal{H}_R$ under certain conditions, these properties are also guaranteed for the VF because of one-to-one correspondence between $\mathcal{H}_R$ and $\mathcal{H}_V$. In the RKHS $\mathcal{H}_R$, an estimate of $R^\phi$ at time instant $n \in \mathbb{Z}_{\geq 0}$ is given by $\hat{R}_n^\phi(x) = \sum_i^r c_i\kappa(x, x_i)$, $c_i \in \mathbb{R}$, $r \in \mathbb{Z}_{\geq 0}$, where $\{x_i\}_{i \in \{1,2,\dots,r\}} \subset \mathcal{X}$ is the set of samples, and the reproducing kernel $\kappa$ is defined in (9). The estimate of the VF $V^\phi$ at time instant $n \in \mathbb{Z}_{\geq 0}$ is then obtained by $\hat{V}_n^\phi(x) = U^{-1}(\hat{R}_n^\phi)(x) = \sum_{i=1}^r c_i K(x, x_i)$, where $K$ is defined in (10). Therefore, when an algorithm promoting sparsity is employed and only fewer kernel functions are employed to estimate the immediate cost function, i.e., $r$ is suppressed small, it immediately implies that sparsity is preserved for the estimate of the VF as well.

## I Experimental settings

We present the experimental settings. The parameter settings for the Mountain Car problem and the simulated inverted pendulum are summarized in Table I.1, and Table I.2, respectively. The parameters were roughly tuned so that the algorithms work reasonably well. However, we conducted no elaborative tuning or heuristic approaches, which are crucial to ensure stable improvements of performance in RL, to further improve performances, because the main purpose of the experiments is to show sensitiveness of DT approaches toward the choice of the time interval.

Table I.1: Summary of the parameter settings for the Mountain Car problem

| Parameters | CTGP | CTKF | GPTD | DTKF |
|---|---|---|---|---|
| Coherence threshold | 0.70 | 0.70 | 0.70 | 0.70 |
| Kernel parameter $\sigma$ | 0.2 | 0.2 | 0.2 | 0.2 |
| Maximum absolute value of control | 1.0 | 1.0 | 1.0 | 1.0 |
| Control cycle | 1.0 (sec) | 1.0 (sec) | 1.0 (sec) | 1.0 (sec) |
| Discount factor $\beta, \gamma$ | $\beta = 0$ | $\beta = 0$ | $\gamma = 1$ | $\gamma = 1$ |
| Standard deviation $\mu_o$ of the observed cost | 0.1 | 0.1 | $0.1\Delta t^2$ | $0.1\Delta t^2$ |
| Cost on controls: Matrix $M$ | $0.001I$ | $0.001I$ | $0.001I$ | $0.001I$ |
| Stochastic term in the SDE: Matrix $A(x, u)$ | 0 | 0 | 0 | 0 |
| Step size | – | 1.8 | – | 0.4 |

Table I.2: Summary of the parameter settings for the simulated inverted pendulum

| Parameters | CTGP | GPTD |
|---|---|---|
| Coherence threshold | 0.95 | 0.95 |
| Kernel parameter $\sigma$ | 0.2 | 0.2 |
| Maximum absolute value of control | 6 | 6 |
| Learning time per update | 10 (sec) | 10 (sec) |
| Time interval $\Delta t$ | 0.01 (sec) | 0.01 (sec) |
| Discount factor $\beta, \gamma$ | $\beta = 0.01$ | $\gamma = e^{-0.01*0.01}$ |
| Standard deviation $\mu_o$ of the observed cost | 0.1 | 0.1 |
| Cost on controls: Matrix $M$ | $0.1I$ | $0.1I$ |
| Stochastic term in the SDE: Matrix $A(x, u)$ | $0.01I$ | $0.01I$ |
| Gravity $g$ | 9.8 | 9.8 |
| Mass $m$ | 1 | 1 |
| Length $\ell$ of pendulum | 1 | 1 |
| Friction effect $\rho$ | 0.01 | 0.01 |

## Footnotes

[1]The first exit time $t_1$ of a bounded set $\mathrm{int}(\mathcal{X})$ is given by $t_1 := \inf\{q \mid x(q) \neq \mathrm{int}(\mathcal{X})\}$ starting from a point $x(0) = x \in \mathrm{int}(\mathcal{X})$.