[Reviews · NeurIPS 2018]

Reviewer 1



Strengths 1. Considering dynamic programming problems in continuous time such that the methodologies and tools of dynamical systems and stochastic di_x000b_eren- tial equations is interesting, and the authors do a good job of motivating the generalities of the problem context. 2. The authors initially describe problems/methods in very general terms, which helps preserve understandability of the _x000c_first sections. Weaknesses 1. The parameterizations considered of the value functions at the end of the day belong to discrete time, due to the need to discretize the SDEs and sample the state-action-reward triples. Given this discrete implementa- tion, and the fact that experimentally the authors run into the conven- tional di_x000e_culties of discrete time algorithms with continuous state-action function approximation, I am a little bewildered as to what the actual bene_x000c_t is of this problem formulation, especially since it requires a re- de_x000c_nition of the value function as one that is compatible with SDEs (eqn. (4) ). That is, the intrinsic theoretical bene_x000c_ts of this perspective are not clear, especially since the main theorem is expressed in terms of RKHS only. 2. In the experiments, the authors mention kernel adaptive _x000c_filters (aka kernel LMS) or Gaussian processes as potential avenues of pursuit for addressing the function estimation in continuous domains. However, these methods are fundamentally limited by their sample complexity bottleneck, i.e., the quadratic complexity in the sample size. There's some experimental ref- erence to forgetting factors, but this issue can be addressed in a rigorous manner that preserves convergence while breaking the bottleneck, see, e.g., A. Koppel, G. Warnell, E. Stump, and A. Ribeiro, Parsimonious on- line learning with kernels via sparse projections in function space," arXiv preprint arXiv:1612.04111, 2016. Simply applying these methods without consideration for the fact that the sample size conceptually is approaching in_x000c_nity, makes an update of the form (16) inapplicable to RL in general. Evaluating the Bellman operator requires computing an expected value. 3. Moreover, the limited complexity of the numerical evaluation is reflective of this complexity bottleneck, in my opinion. There are far more effective RKHS value function estimation methods than GPTD in terms of value function estimation quality and memory efficiency: A. Koppel, G. Warnell, E. Stump, P. Stone, and A. Ribeiro. Policy Evaluation in Continuous MDPs with E_x000e_cient Kernelized Gradient Tem- poral Di_x000b_fference," in IEEE Trans. Automatic Control (submitted), Dec. 2017." It's strange that the authors only compare against a mediocre benchmark rather than the state of the art. 4. The discussion at the beginning of section 3 doesn't make sense or is written in a somewhat self-contradictory manner. The authors should take greater care to explain the di_x000b_erence between value function estimation challenges due to unobservability, and value function estimation problems that come up directly from trying to solve Bellman's evaluation equation. I'm not sure what is meant in this discussion. 5. Also, regarding L87-88: value function estimation is NOT akin supervised learning unless one does Monte Carlo rollouts to do empirical approxima- tions of one of the expectations, due to the double sampling problem, as discussed in R. S. Sutton, H. R. Maei, and C. Szepesvari, \A convergent o(n) temporal- di_x000b_erence algorithm for o_x000b_-policy learning with linear function approxi- mation," in Advances in neural information processing systems, 2009, pp. 1609?1616. and analyzed in great detail in : V. R. Konda and J. N. Tsitsiklis, Convergence rate of linear two-timescale stochastic approximation," Annals of applied probability, pp. 796- 819, 2004. 6. The Algorithm 1 pseudo-code is strangely broad so as to be hand-waving. There's no speci_x000c_cs of a method that could actually be implemented, or even computed in the abstract. Algorithm 1 could just as well say "train a deep network" in the inner loop of an algorithm, which is unacceptable, and not how pseudo-code works. Specifically, one can't simply "choose" at random" an RKHS function estimation algorithm and plug it in and assume it works, since the lion-share of methods for doing so either re- quire in_x000c_nite memory in the limit or employ memory-reduction that cause divergence. 7. L107-114 seems speculative or overly opinionated. This should be stated as a remark, or an aside in a Discussion section, or removed. 8. A general comment: there are no transitions between sections, which is not good for readability. 9. Again, the experiments are overly limited so as to not be convincing. GPTD is a very simplistic algorithm which is not even guaranteed to pre- serve posterior consistency, aka it is a divergent Bayesian method. There- fore, it seems like a straw man comparison. And this comparison is con- ducted on a synthetic example, whereas most RL works at least consider a rudimentary OpenAI problem such as Mountain car, if not a real robotics, power systems, or _x000c_financial application.

Reviewer 2



The authors introduce a strategy for continuous-time value function approximation based on RKHS. Thus, an operator is proposed to relate two different RKHSs aiming to find a one-to-one correspondence between unobservable and observable output spaces. As seen in the theoretical analysis, unobservable vectors can be estimated as a convex combination of kernels. Also, barrier-certified policy updates are coupled through quadratic programming. Practical validation includes the inclusion of GPs and kernel adaptive filters to model the observable outputs, testing them on an inverted pendulum and a planer point-mass simulations. The theoretical derivations are complemented in the supplementary material (derivations in the paper are difficult to follow and/or incomplete in different sections), especially, those related to the practical implementation using a Gaussian kernel. The experimental results could be enhanced to make clear the virtues of the paper. So, a suitable comparison with state-of-art approaches could be included.

Reviewer 3



This paper presents an approach for estimating value functions of a given policy in a stochastic continuous-time setting. The approach is based on the theory of RKHS. Specifically, the authors construct an invertible map U between the instantaneous reward function and the underlying RKHS that the value function lives in. The authors then proposed to learn the instantaneous reward function via supervised learning, and then pass the estimated reward function through U^{-1} to obtain the estimated value function. The authors provide a theoretically sound construction of this map. While I have some concerns about the applicability and usefulness of the authors proposed method in practice, the theoretical construction given is interesting and hence I believe this paper should be accepted. Below are my comments: 1. What is meant in the introduction when stated that "CT formulations are also advantageous for control-theoretic analyses such as stability and forward invariance"? Both these notions of stability and invariance make sense in discrete time as well. 2. In practice, even when the system operates in continuous time, one typically implements the controller digitally. Hence, one still need to deal with discretization effects; one can only for instance apply a constant input throughout a small time step (e.g. a zero-order hold). It is not obvious to me that designing a continuous time controller, and then implementing its discretized version, is more or less preferable to directly designing a discrete controller. 3. Please state what exactly is assumed to be known to the algorithm and what is unknown. For instance, in equation (1), is the drift function h and the diffusion function eta known? I believe it must be known, because the mapping U(.) contains these terms. On the other hand, I believe that the cost function R(.) is assumed to be not known, and learned from data. A more common setting in RL is where one assumes that the description of the dynamics is *not* available to the algorithm, and one must learn in this setting. For instance, the GPTD paper of Engel et al. works in a model free manner. 4. Let us say one knows the reward function (a reasonable assumption in practice). Does Theorem 1 then imply one can compute a solution to the HJB equation (5) by simply evaluating U^{-1}(R(.))? Can you show with a simple calculation what happens when the dynamics and cost is given by something where the analytical solution is know, e.g. LQR? In LQR, we know that both the R(.) and the value function are quadratics. Having this calculation would be very illuminating in my opinion. 5. In Theorem 1, what happens if one has a policy that has multiple equilibrium points, e.g. different values of starting points x in X can convergence to different points? A unique equilibrium point seems like a possibly restriction assumption for non-linear systems. 6. In Figure 3, can you comment on why your method CTGP appears to have a lot more variance than the existing GPTD method? Also, how is the policy initialized and updated for the inverted pendulum experiment?